# In Vivo Hepatoprotective and Nephroprotective Activity of Acylated Iridoid Glycosides from *Scrophularia hepericifolia*

**DOI:** 10.3390/biology10020145

**Published:** 2021-02-12

**Authors:** Maged S. Abdel-Kader, Saleh I. Alqasoumi

**Affiliations:** 1Department of Pharmacognosy, College of Pharmacy, Prince Sattam Bin Abdulaziz University, Al-Kharj 11942, Saudi Arabia; 2Department of Pharmacognosy, College of Pharmacy, Alexandria University, Alexandria 21215, Egypt; 3Department of Pharmacognosy, College of Pharmacy, King Saud University, Riyadh 11451, Saudi Arabia; sqasoumi@ksu.edu.sa

**Keywords:** *Scrophularia hypericifolia*, acylate iridoid glycoside, hepatoprotective, nephroprotective

## Abstract

**Simple Summary:**

Chronic liver disease is a major life-threating problem worldwide. Deaths due to liver disease constituted 2.4% of total deaths globally in 2017. Yet most of the available drugs for improving liver conditions are of natural origin. More research is necessary to discover agents that are more effective. Previously, the extract of *Scrophularia hypericifolia* showed promising protection of liver and kidney tissues against induced toxicity in experimental animals. The current phytochemical study aimed to identify the active molecules in the extract. Nine iridoid glycoside derivatives were identified from the plant extract, including four new compounds after extensive chromatographic purification. The structures were identified by applying various spectroscopic methods. Biological evaluation for protective effect on liver and kidney tissues was conducted on compounds isolated with a high enough yield. Serum and tissue parameters as well as histopathological studies were conducted for efficacy evaluation. Two of the new compounds showed the best protection of liver and kidney tissues as indicated by the studied parameters. These findings indicated that natural products could provide solutions to health problems. The role of nutraceuticals in managing liver problems is a promising field for further studies.

**Abstract:**

Phytochemical investigation of the chloroform fraction obtained from *Scrophularia hypericifolia* aerial parts led to the isolation of nine acylated iridoid glycosides. The new compounds were identified as 6-O-α-L(2″-acetyl, 3″,4″-di-O-*trans*-cinnamoyl) rhamnopyranosyl-6′-acetyl catalpol (6′-acetyl hypericifolin A) (**1**), 6-O-α-L(2″, 4″-diacetyl, 3″-O-*trans*-cinnamoyl) rhamnopyranosyl-6′-acetyl catalpol (6′-acetyl hypericifolin B) (**2**), 6-O-α-L(2″-acetyl, 3″,4″-di-O-*trans*-cinnamoyl) rhamnopyranosyl catalpol (hypericifolin A) (**3**) and 6-O-α-L(2″, 4″-diacetyl, 3″-O-*trans*-cinnamoyl) rhamnopyranosyl catalpol (hypericifolin B) (**4**). Previously reported compounds were identified as laterioside (**5**), 8-O-acetylharpagide (**6**), 6-O-α-L(4′-O-*trans*-cinnamoyl) rhamnopyranosyl catalpol (**7**), lagotisoside D (**8**) and harpagoside (**9**). Identification achieved via analyses of physical and spectral data including 1D, 2D NMR and High Resolution Electrospray Ionization Mass spectroscopy (HRESIMS). Compounds **2**–**4** and **6** were subjected to biological evaluation against paracetamol-induced toxicity. The biochemical parameters aspartate aminotransferase (AST), alanine aminotransferase (ALT), alkaline phosphatase (ALP) and gamma glutamyl transpeptidase (GGT) as well as total bilirubin were used to access the liver condition. Measurement of serum levels of urea, creatinine, sodium and potassium cations were indicators for kidney condition. Liver and kidney samples were subjected to histopathological study. The best protection was found in the group treated with **3** followed by **4** and **6**, while **2** was almost inactive.

## 1. Introduction

The genus *Scrophularia* consists of 200–300 species, present mainly in Asia, Europe and, to a lesser extent, North America [1]. Only four species of the genus *Scrophularia* are present in Saudi Arabia [2,3]. Members of this genus are used traditionally as anti-inflammatory, anticancer, laxative, diuretic, heart and circulatory stimulant treatments [4,5]. They are also applied for the treatment of pharyngitis, neuritis, laryngitis, smallpox, measles, high-heat plague and poisoning, dermatitis and rheumatoid arthritis [4,6,7]. Pharmacological studies revealed that members of the genus possess antibacterial, antioxidant [8], anti-inflammatory, antidiabetic [9], cardioprotective [10], hepatoprotective, immunostimulant [11,12], wound healing [13] and insecticidal [14] activities.

*Scrophularia* species are rich in iridoid glycosides. The hepatoprotective effect of *S. koelzii* was traced to the iridoid glycosides scropolioside A, harpagoside and koelzioside [12]. Scropolioside D2 and harpagoside B isolated from *S. deserti* showed significant anti-inflammatory and antidiabetic effects [9]. Iridoid glycosides from *S. umbrosa* were reported to have inhibitory activity on nitric oxide production [15]. Two iridoid glycosides with analgesic activity were reported from *S. kotscyhana* [16].

Chronic liver diseases are one of the major causes of morbidity and mortality globally. In 2017, liver diseases caused more than 1.32 million deaths in females and 883,000 in males, compared with less than 899,000 deaths in 1990 worldwide. Deaths due to liver disease constituted 2.4% of total deaths globally in 2017 compared with 1.9% in 1990 [17]. Non-alcoholic fatty liver disease (NAFLD) is characterized by inflammation and progressive tissue degeneration. It affects about 5% of the general adult population, and 20% of obese people [18]. In the absence of specific approved drugs on the market for the treatment of NAFLD, some nutraceuticals could contribute to the improvement of lipid infiltration and/or biochemical parameters. A few nutraceuticals were studied clinically for their effects against NAFLD, including silymarin (Sil), vitamins E and D, curcumin, resveratrol and the total extracts of *Salvia milthiorriza* [19].

*Scrophularia hypericifolia* growing in Saudi Arabia was reported to have hepatoprotective and nephroprotective activities [20]. In the current study, the secondary metabolites of the plant were isolated and identified by various spectroscopic methods. Pure compounds isolated in good yield were evaluated for their hepatoprotective and nephroprotective activities.

## 2. Results and Discussion

### 2.1. Structure Elucidation

The total extract of the aerial parts of *S. hypericifolia* was partitioned by liquid–liquid fractionation to a petroleum ether-soluble fraction, chloroform-soluble fraction and n-butanol-soluble fraction. Extensive chromatographic fractionation and purification of the chloroform-soluble fraction resulted in the isolation of compounds **1**–**9** (Figure 1) in addition to cinnamic acid. The known compounds were identified by comparison of their measured spectra with the literature data. All the assignments of ^1^H and ^13^C values were based on DEPT135, COSY, HSQC, HMBC, H2BC, NOESY and/or ROESY experiments. Compound **5** was identified as laterioside, an iridoid glycoside first identified from *S. lateriflora* [21] then reported from *S. deserti* [22] as well as many *Verbascum* species [23]. Compound **6** was identified as 8-O-acetylharpagide common in *Scrophularia* species [24]. 8-O-Acetylharpagide is reported to have antibacterial, anti-inflammatory, vasoconstrictor and cancer chemopreventive activities [25,26,27,28]. Compound **7** was identified as 6-O-α-L(4″-O-*trans*-cinnamoyl) rhamnopyranosyl catalpol. Compound **7** is reported in this work for the first time from *Scrophularia* species. Previously, it was characterized from *Gmelina arborea* (Lamiaceae) and *Buddleja polystachya* (Scrophulariaceae) [29,30]. The data of **8** enable the identification of the structure as lagotisoside D previously isolated from *Lagotis yunnanensis* and *S. dentata* [31,32]. Compound **9** data were identical to those reported for harpagoside [33]. Harpagoside was first isolated from *Harpagophytum procumbens* F. Pedaliaceae (devil’s claw) [34]. Harpagoside proved to have potent anti-rheumatic, anti-inflammatory and analgesic effects [35].

The HRESIMS of **1** showed an [M+Na]^+^ m/z of 875.2733 and [M+K]^+^ m/z of 891.2471 for the molecular formula C_43_H_48_O_18_ (Appendix A). The ^1^H and ^13^C NMR data of **1** (Table 1 and Table 2) indicated the presence of two acetyl moieties at δ_H_ 2.07, δ_C_ 19.54, 171.40; δ_H_ 2.19, δ_C_ 19.48, 170.32 and two *trans*-cinnamoyl moieties as indicated by two conjugated ester carbonyls at δ_C_ 165.93, 166.31 and the *trans*-coupled protons at δ_H_ 6.40, 6.50, 7.61, 7.70 (each d, *J* = 16 Hz) along with the overlapped aromatic protons at δ_H_ 7.35, 7.49 and 7.53 (Appendix A). ^13^C NMR of **1** showed ten carbon resonances sorted by DEPT 135 experiments into 7 X CH including two oxygenated, two vinyls and one di-oxygenated methins, one CH_2_O and one oxygenated quaternary carbon. Such an array (Table 1 and Table 2) pointed to an iridoid skeleton of a catalpol skeleton [36]. Another set of carbons, including four CHO, CH_2_O and one di-oxygenated CH were assigned to glucose moiety attached to C-1 of the iridoid aglycone. The relative stereochemistry of the iridoid aglycone was confirmed from the NOE correlations between the protons H-1 and H-6; H-6 and H-7; H-6 and H-10, all with α-orientation (Appendix A). The chemical shift of C-6 at δ_C_ 83.74 was diagnostic for a substituted position. The six carbon resonances at δ_C_ 96.41, 71.05, 69.92, 69.45, 66.89 in addition to the methyl carbon at δ_C_ 16.53 correlated by the HSQC experiment with the proton doublets at δ_C_ 1.27 in ^1^H NMR spectra were assigned to a rhamnosyl substituent (Appendix A). HMBC correlations between the H-6 doublet at δ_H_ 4.03 and C-1″ of rhamnose at δ_C_ 96.41δ_C_ as well as the NOE correlation between H-6 and H-1″ singlet at δ_H_ 5.13 all support the position of a rhamnosy moeity at C-6. Many iridoid glycosides including **7** and **8** bearing a rhamnose moiety at C-6 were previously reported (Appendix A) [29,30,31,32]. Compared with compound **7**, **1** bears an additional two acetyls and one *trans*-cinnamoyl (Table 1 and Table 2). The H-6′ and C-6′ of the glucose moiety in **1** were downfield shifted to δ_H_ 4.28 (dd, *J =* 5, 12 Hz), 4.47(d, *J =* 6.8 Hz) and δ_C_ 62.74 compared with normal values of glucose [33], indicating acylation at that position. NOE correlations were observed between the acetyl protons at δ_H_ 2.07 (s) and H-6′ at δ_H_ 4.28 (dd, *J =* 5, 12 Hz), 4.47(d, *J =* 6.8 Hz), H-5′ at δ_H_ 3.52 (m) and H-4′ at 3.43 (m) (Appendix A). Further evidence for C-6′ acetylation was obtained from the HMBC correlations between both H-6′ protons, and the methyl protons at δ_H_ 2.07 with the acetyl carbonyl at δ_C_ 171.40 (Appendix A). The H2BC experiment was extensively used to assign protons and carbons, especially those of the rhamnose moiety (Appendix A). The chemical shifts of H-2″, H-3″, H-4″ at δ_H_ 5.43 (bs), 5.53 (dd, *J =* 2.7, 10 Hz) and 5.32 (t, *J =* 10 Hz) indicated that the three positions are acylated. The assignments of the two cinnamoyl and the left acetyl substituents to their exact positions on C-2″-C4″ were achieved via detailed analyses of the HMBC correlations. The broad singlet assigned for H-2″ at δ_H_ 5.43 as well as the acetyl methyl singlet at δ_H_ 2.19 showed three bond correlations with the acetyl carbonyl at δ_C_ 170.32. HMBC correlations were observed between H-3″ at 5.53 (dd, *J =* 2.7, 10 Hz), the two *trans*-oriented cinnamoyl doublets at δ_H_ 6.40 and 7.61 and the carbonyl signal at δ_C_ 165.93 assigned to C-9‴, while the H-4″ triplet at δ_H_ 5.32, and the two *trans*-oriented cinnamoyl doublets at δ_H_ 6.50 and 7.70 were correlated to the carbonyl signal at δ_C_ 166.31 assigned to C-9″″ (Appendix A). Based on the provided evidence, **1** was identified as 6-O-α-L(2″-acetyl, 3″,4″-di-O-trans-cinnamoyl) rhamnopyranosyl-6′-acetyl catalpol and was given the name 6′-acetyl hypericifolioside A.

The HRESIMS of **3** showed an [M+Na]^+^ m/z of 833.2616 and [M+K]^+^ m/z of 849.2348 for the molecular formula C_41_H_46_O_17_, indicating one acetyl group less than **1** (Appendix A). The appearance of H-6′ and C-6′ of glucose at δ_H_ 3.67 (dd, *J =* 6.5, 11.9 Hz), 3.96 (d, *J =* 10.9 Hz) and δ_C_ 61.58 in **3** indicated that C-6′ hydroxyl is free (Appendix A). Spectra of **3** were repeated in pyridine-d6 to resolve overlapping peaks in the CD_3_OD spectra, such as H-1, H-4 and H-1″ (Table 1) (Appendix A). The assignment of the two cinnamoyl moieties at 3″, 4″ and the acetyl group at 2″ of the rhamnose moiety followed the same spectral evidence explained for **1**. The broad singlet assigned for H-2″ at δ_H_ 5.43 as well as the acetyl methyl singlet at δ_H_ 2.19 showed three bond correlations with the acetyl carbonyl at δ_C_ 170.31. HMBC correlations were observed between H-3″ at δ_H_ 5.55 (dd, *J =* 3.2, 10.2 Hz), the two *trans*-oriented cinnamoyl doublets at δ_H_ 6.39 and 7.60 and the carbonyl signal at δ_C_ 165.89 assigned to C-9‴, while the H-4″ triplet at δ_H_ 5.33 and the two *trans*-oriented cinnamoyl doublets at δ_H_ 6.49 and 7.71 correlated to the carbonyl signal at δ_C_ 166.30 assigned to C-9‴ (Appendix A). Consequently, **3** was identified as 6-O-α-L(2″-acetyl, 3″,4″-di-O-*trans*-cinnamoyl) rhamnopyranosyl catalpol and was given the trivial name hypericifolioside A.

Similar to **1** and **3**, the spectra of **2** pointed out a catalpol skeleton with a C-6 rhamnosyl substituent. The ^1^H and ^13^C NMR data of **2** (Table 1 and Table 2) (Appendix A) indicated the presence of three acetyl moieties at δ_H_ 2.00, δ_C_ 20.94, 170.72; δ_H_ 2.02, δ_C_ 21.07, 170.32; δ_H_ 2.13, δ_C_ 21.04, 170.06 and one *trans*-cinnamoyl moiety, as indicated by the conjugated ester carbonyls at δ_C_ 165.75 and the *trans*-coupled protons at δ_H_ 6.54, 7.60 (each d, *J =* 16 Hz) along with the aromatic protons at δ_H_ 7.43, 7.69. HRESIMS of **2** were in full support of the suggested structure, where it showed an [M+Na]^+^ m/z of 787.2412 and [M-1]^−^ m/z of 763.2454 for the molecular formula C_36_H_44_O_18_ (Appendix A). This spectral evidence indicated that **2** has three acetyl groups over the structure of **7**. The H-6′ and C-6′ of glucose were downfield shifted to δ_H_ 4.14 (m), 4.31(m); δ_C_ 63.27 (DMSO), δ_H_ 4.69 (dd, *J =* 3.7, 11.4 Hz), 4.88 (d, *J =* 11.4 Hz) and δ_C_ 63.61 (pyridine-d6) (Appendix A) compared with normal values of glucose [34]. This downfield shift supports C-6′ acetylation. More evidence for C-6′ acetylation was obtained from NOESY experiments where NOE correlations were observed between the acetyl protons at δ_H_ 2.00 (s) and H-5′ at δ_H_ 3.99 (m) (Appendix A). An H2BC experiment was applied to assign the rhamnose protons and carbons (Appendix A). The downfield shift of H-2″, H-3″, H-4″ to δ_H_ 5.91 (s), 5.95 (d, *J =* 7.7 Hz) and 5.71 (t, *J =* 9.6 Hz) (pyridine-d6) (Appendix A) pointed out their acylated nature. The position of the cinnamoyl moiety was decided through the analysis of HMBC experiment correlations measured in pyridine d6 (Appendix A). Three bond contours were clear between H-3″ at δ_H_ 5.95 and the cinnamoyl carbonyl signal at δ_C_ 165.89. Both H-2″, H-4″ at δ_H_ 5.91, 5.71 showed three bond correlations with the two acetyl carbonyls at δ_C_ 169.89 and 170.21 (Appendix A). Based on the above discussion, compound **2** was identified as 6-O-α-L (2″, 4″-diacetyl, 3″-O-*trans*-cinnamoyl) rhamnopyranosyl-6′-acetyl catalpol and was given the trivial name 6′-acetyl hypericifolioside B.

HRESIMS data of **4** were consistent with the molecular formula C_34_H_42_O_17_, as shown from the [M+Na]^+^ at m/z of 745.2313, [M+K]^+^ at m/z of 761.2051 and [M−1]^−^ at m/z of 721.2355 (Appendix A). The absence of one acetyl group was clear in the NMR data and fully supported by the MS data (Appendix A). In the spectra of **4**, the appearance of H-6′ and C-6′ of glucose at δ_H_ 3.65 (dd, *J =* 6.8, 12 Hz), 3.94 (d, *J =* 12 Hz) and δ_C_ 61.55 indicated that **4** is the C-6′ deacetylation derivative of **2**. Similar to **1**–**3**, the acylations in **4** were located at H-2″ (δ_H_ 5.88, s), H-3″ (δ_H_ 5.94, bd, *J =* 9.7 Hz) and H-4″ (δ_H_ 5.71, bd, *J =* 9.7 Hz) based on their downfield shift values. The *trans*-cinnamoyl moiety was assigned to position 3″ based on the HMBC correlation between H-3″ at δ_H_ 5.94 and the cinnamoyl carbonyl at δ_C_ 165.88 (Appendix A). Compound **4** was identified as 6-O-α-L(2″, 4″-diacetyl, 3″-O-*trans*-cinnamoyl) rhamnopyranosyl catalpol and was given the trivial name hypericifolioside B.

### 2.2. Biological Evaluation

The total extract of *S. hypericifolia* showed promising hepatoprotective and nephroprotective activities [20]. Compounds **2**–**4** and **6** isolated in good yield were subjected to biological testing against paracetamol (Pa)-induced liver and kidney toxicities. Toxic doses of Pa produce fatal hepatic necrosis in humans and other mammals, including rats and mice [37]. Toxic doses of Pa cause saturation of the sulfation and glucoronidation routes of metabolism. As an alternative to get rid of the extra Pa, the cytochrome P450 enzymes are enhanced to oxidize a higher percentage of Pa molecules to the highly reactive N-acetyl-p-benzoquinone imine (NAPQI) species. NAPQI’s loss of one electron results in the formation of semi-quinone radicals with an extremely short half-life. It is then rapidly conjugated with the sulphydryl donor glutathione (GSH), resulting in the depletion of the liver GSH pool [38]. Excessive formation of NAPQI as well as glutathione store depletion leads to covalent binding of NAPQI to vital proteins and the lipid bilayer of hepatocyte membranes and enhances lipid peroxidation. These consequences lead to hepatocellular death and centrilobular liver necrosis [39]. The transport system of the hepatocytes was impaired, leading to the leakage of plasma membrane [40], reflected by increases in serum enzyme levels. Treatment of animals with Pa resulted in a dramatic elevation of transaminases (aspartate aminotransferase (AST), alanine aminotransferase (ALT)) and alkaline phosphatase (ALP) levels. Severe jaundice is expressed by the elevation of serum bilirubin levels (Figure 2 and Appendix A) [41].

#### 2.2.1. Hepatoprotective Effect

Administration of Sil, at a dose of 10 mg/kg (20.7 μmol/kg) prior to Pa resulted in a significant correction (*p* < 0.001) in the elevated AST (37.74%), ALT (43.29%), gamma glutamyl transpeptidase (GGT) (37.53%), ALP (27.31%) and bilirubin (54.15%) levels in the corresponding group of rats (Figure 2 and Appendix A). Sil acts by several mechanisms including an antioxidant effect by scavenging prooxidant free radicals and via restoring the concentration of GSH. Sil also restores the normal cellular membrane function, resulting in protection against xenobiotic injury. Sil also initiates the synthesis of ribosomal RNA via activation of DNA polymerase-I and steroid-like action in regulating DNA transcription and enhancement of protein synthesis necessary for the regeneration of liver cells [42,43]. 

Treatment of rats with **3** at 20.7 μmol/kg doses prior to Pa showed a significant (*p* < 0.01; 0.001) reduction by 23.26, 33.71, 37.95, 16.67 and 27.70% in the elevated levels of AST, ALT, GGT, ALP and bilirubin. Compound **4** showed less protection, expressed as 13.86, 26.72, 36.14, 13.99 and 25.00% reductions in the levels of AST, ALT, GGT, ALP and bilirubin. Compound **6** showed weaker effects on the serum biochemical parameters, while **2** was almost inactive (Figure 2). The impact of the tested compounds was also evaluated on total protein (TP) and non-protein sulfhydryl groups (NP-SH) levels in liver cells (Figure 3A, Figure 4 and Appendix A). Compound **3** restored TP contents to about 50% of that of Sil. The effect of **3** restoring NP-SH (3.43 ± 0.30) was slightly less than the standard drug Sil (3.37 ± 0.28). The effects of **4** were less than **3**, followed by **6**. The results of the histopathological study were in support of the serum biochemical and tissue parameters obtained. Compared with the normal hepatocytes (Figure 5A), the liver samples of the group only treated with Pa (Figure 5B) showed severe damage, expressed as portal vessel congestion, necrosis and infiltration. The Sil-treated group indicated that Sil restores the liver cell architecture closer to the normal state (Figure 5C) with little congestion. The group treated with **3** expressed a great level of protection (Figure 5D) where the appearance of cells was almost normal. Mild focal necrosis and portal tract congestion were observed in the liver specimen of the group treated with **4** (Figure 5E), indicating a moderate level of protection. Less protection was evident from treatment with **6** as the cells suffered more from degeneration, portal congestion and inflammatory cell infiltration.

#### 2.2.2. Nephroprotective Effect

The kidney is a vital organ responsible for the regulation of plasma ionic composition and the elimination of nitrogenous end metabolic waste products such as urea, creatinine and uric acid [44]. Elevations of these parameters are reliable parameters for nephrotoxicity [45]. Pa nephrotoxicity is reflected by elevated serum levels of creatinine, urea, sodium and potassium (Appendix A). The levels of NP-SH and TP were diminished (Appendix A). Treatment with Sil resulted in marked improvement of all these parameters. The group treated with **3** showed the most effective degree of protection among the tested compounds. Prior treatment with Sil resulted in a 39.87% reduction in urea, a 49.88% reduction in creatinine, a 57.46% reduction in sodium and a 46.03% reduction in potassium levels compared with 24.61%, 36.00%, 33.88% and 35.47% reductions in the same parameters, respectively, as a result of the administration of **3** (Figure 6, Appendix A). Compound **3** was as effective as Sil in restoring NP-SH groups and slightly less in restoring TP (Figure 3B and Figure 4, and Appendix A). Less improvement was observed with treatment with **4**, followed by **6**, while **2** was totally ineffective. The normal kidney cells structure is presented in Figure 7A. Histopathological study revealed that the kidney cells of rats treated with Pa expressed severe changes (Figure 7B), such as degenerated corpuscle, reduced size of glomeruli, shrinkage of Bowman capsule, cloudy swelling in tubules and inflammatory infiltrate. Pre-administration of Sil (Figure 7C) resulted in nearly normal glomeruli and mild cloudy swelling in tubules, while the group treated with **3** (Figure 7D) expressed moderate cortical atrophy and moderate reduction in glomerular size. More damage was observed in groups treated with **4** (Figure 7E) and **6** (Figure 7F).

Many nutraceuticals containing well-known traditional plants are utilized for their liver and/or kidney health benefits. Extracts of *Nigella sativa* [46], *Chenopodium ambrosioides* [47], *Hemidesmus indicus* [48], *Bryophyllum pinnatum* [49] and *Verbascum thapsus* [50] are reported to have hepatoprotective and/or nephroprotective effects. Interestingly, *Verbascum thapsus* is rich in iridoid contents [50]. One of its components, the iridoid glycoside aucubin, was reported to have wide range of pharmacological activities, including hepatoprotective and antioxidant effects [51]. The hepatoprotective effects of related iridoid glycosides were reviewed [52]. Molecular docking analysis was performed to assess the binding efficiencies of iridoids to the active sites of human CYP3A4 enzyme. Selected iridoids were evaluated for their hepatoprotective effects using quantitative real-time PCR to measure the expression levels of CYP3A4 mRNA in HepG2 cells. The tested iridoids showed an inductive effect on CYP3A4 mRNA levels in the HepG2 cells. The study results indicated that the hepatoprotective effects were caused by the facilitation of drug metabolism and the reduction of both mitochondrial dysfunction and oxidative stress [53].

## 3. Materials and Methods

### 3.1. General Experimental Procedures

Melting points were determined in open capillary tubes using a Thermosystem FP800 Mettler FP80 central processor supplied with an FP81 MBC cell apparatus and are shown uncorrected. Ultraviolet absorption spectra were measured on a Unicum Heyios a UV–Visible spectrophotometer. A Jasco P-2000 Polarimeter was used to measure the optical rotations. ^1^H, ^13^C-NMR and 2D-NMR data were measured on a Bruker UltraShield Plus 500 MHz spectrometer at NMR Unite, College of Pharmacy, Prince Sattam Bin Abdulaziz University, operated at 500 MHz for protons and 125 MHz for carbon atoms. Chemical shift values were reported in δ (ppm) relative to the residual solvent peaks. Coupling constants (*J*) are reported in Hertz (Hz). 2D-NMR experiments (COSY, HSQC, HMBC, H2BC, NOESY and/or ROESY) were performed utilizing the standard Bruker program. HRMS were determined by direct injection using a Thermo Scientific UPLC RS Ultimate 3000 Q Exactive hybrid quadrupole—Orbitrap mass spectrometer combining high-performance quadrupole precursor selection with high-resolution, accurate mass (HR/AM) Orbitrap™ detection. Direct infusion of isocratic elution acetonitrile/methanol (70:30) with 0.1% formic acid was used to flush the samples. Runtime was 1 min using nitrogen as auxiliary gas with a flow rate of 5 μL/min. A scan range from 160–1500 m/z was used. Resolving power was adjusted to 70,000 @ m/z 200. Detection was in both positive and negative modes separately. Calibration was done using Thermo Scientific Pierce™ LTQ Velos ESI Positive Ion Calibration Solution including caffeine, Met-Arg-Phe-Ala (MRFA), Ultramark 1621, n-Butyl-amine components and Pierce™ LTQ Velos ESI Negative Ion Calibration Solution including sodium dodecyl sulphate (SDS), sodium taurocholate and Ultramark 1621 components. The capillary temperature was set at 320 °C and capillary voltage at 4.2 Kv. A Sephadex LH-20 (Amersham Biosciences, Uppsala, Sweden), silica gel 60/230–400 mesh (EM Science) and RP18 silica gel 40–63/230–400 mesh (Fluka) were used for column chromatography. Centrifugal preparative thin layer chromatography (CPTLC) using a 2 mm silica gel P254 disc was performed on a Chromatotron (Harrison Research Inc. model 7924, San Bruno, CA, USA). The thin layer chromatography (TLC) analysis was performed on Kiesel gel 60 F254 and RP-18 F254S (Merck) plates. A UV lamp (entela Model UVGL-25) operated at 254 nm was used for detecting spots on the TLC plates.

### 3.2. Plant Material

The plants of *Scrophularia hypericifolia* Wydler (aerial parts) were described earlier [20] and collected in March, 2019 from Al-Qassim Province, Saudi Arabia. The plants were identical to the voucher specimen (# 13274) preserved at the herbarium of the Medicinal, Aromatic and Poisonous Plants Research Center (MAPPRC).

### 3.3. Extraction and Isolation

Air-dried powdered aerial parts (1000 g) were repeatedly extracted with 95% ethanol by percolation at room temperature till exhaustion. The solvent was distilled off under reduced pressure using a rotary vacuum evaporator at 40 °C to give 130 g residue. Part of the obtained dried extract (35 g) was dissolved in 800 mL of 40% aqueous ethanol and fractionated with light petroleum (500 mL × 3) to yield 9.25 g of light petroleum-soluble fraction, chloroform (500 mL × 4) to yield 15.32 g of chloroform-soluble fraction and n-butanol (400 mL × 2) to yield 3.15 g of n-butanol-soluble fraction.

The chloroform-soluble fraction was fractionated chromatographically using a silica gel column (150 × 7 cm i.d., 450 g) eluting with chloroform, followed by chloroform/methanol mixtures with a gradual increase in methanol content. Fractions of 200 mL each were collected, screened by TLC and similar fractions were pooled to yield 6 major fractions (A–F). Fraction A (1.51 g) eluted with 5% methanol in chloroform was further purified using an RP18 column (45 × 2 cm i.d., 40 g) eluting with 20% water in methanol to afford 85 mg of **1** and 328 mg of **2** after crystallization from methanol. Fraction B (0.93 g) eluted with 10% methanol in chloroform afforded 325 mg of **3** after purification on an RP18 column (45 × 2 cm i.d., 30 g) eluting with 20% water in methanol. Fraction C (2.73 g) eluted with 10% methanol in chloroform was chromatographed on an RP18 column (90 × 2.5 cm i.d., 90 g) eluting with 20% water in methanol to afford another 200 mg of **3**. Further purification on a Sephadex LH20 eluting with 10% water in methanol afforded 272 mg of **4** from fractions 4–7. Fractions 11–14 were further purified using a Chromatotron equipped with 2 mm silica gel plates and 2% methanol in acetonitrile as the mobile phase to afford 17 mg of cinnamic acid and 92 mg of **5**. Fraction D (1.43 g) eluted with 10% methanol in chloroform was chromatographed on an RP18 column (90 × 2.5 cm i.d., 45 g) eluting with 20% water in methanol to afford 439 mg of **6**. Fraction E (0.67 g) eluted with 20% methanol in chloroform was further purified on a Sephadex LH20 eluting with 20% water in methanol followed by separation on a Chromatotron equipped with 2 mm silica gel plates and 15% methanol in chloroform to afford 32 mg of **7** and 58 mg of **8**. Fraction F (0.43 g) eluted with 20% methanol in chloroform was chromatographed on an RP18 column (45 × 2 cm i.d., 90 g) eluting with 30% water in methanol to afford another 130 mg of **9**.

### 3.4. Compound Characterization

#### 3.4.1. 6′-Acetyl hypericifolioside A (1)

White powder; m.p. 120.8 °C, [α]_25_^D^ -58; UV λ_max_ MeOH: 220, 302, 325 nm; ^1^H and ^13^C NMR see Table 1 and Table 2; HRESIMS [M+Na]^+^ m/z 875.2733 (calcd for C_43_H_48_O_18_Na, 875.2733), [M+K]^+^ m/z 891.2471 (calcd for C_43_H_48_O_18_K, 891.2478).

#### 3.4.2. 6′-Acetyl hypericifolioside B (2)

White powder; m.p. 126.9 °C, [α]_25_^D^ -155; UV λ_max_ MeOH: 218, 301, 326 nm; ^1^H and ^13^C NMR see Table 1 and Table 2; HRESIMS [M+Na]^+^ m/z 787.2412 (calcd for C_36_H_44_O_18_Na, 787.2425), [M-1]^−^ m/z 763.2454 (calcd for C_36_H_43_O_18_, 763.2449).

#### 3.4.3. Hypericifolioside A (3)

White powder; m.p. 138.0 °C, [α]_25_^D^ -56; UV λ_max_ MeOH: 220, 300, 326 nm; ^1^H and ^13^C NMR see Table 1 and Table 2; HRESIMS [M+Na]^+^ m/z 833.2616 (calcd for C_41_H_46_O_17_Na, 833.2633), [M+K]^+^ m/z 849.2348 (calcd for C_41_H_46_O_17_K, 849.2372).

#### 3.4.4. Hypericifolioside B (4)

White powder; m.p. 133.4 °C, [α]_25_^D^ -154; UV λ_max_ MeOH: 221, 301, 325 nm; ^1^H and ^13^C NMR see Table 1 and Table 2; HRESIMS [M+Na]^+^ m/z 745.2313 (calcd for C_34_H_42_O_17_Na, 745.2320), [M+K]^+^ m/z 761.2051 (calcd for C_34_H_42_O_17_k, 761.2059), [M−1]^−^ m/z 721.2355 (calcd for C_34_H_41_O_17_, 721.2344).

### 3.5. Animals

Male Wistar albino rats of average weight (160–180 g) (age 8–10 weeks) were secured by the Experimental Animal Care Center, College of Pharmacy, Prince Sattam Bin Abdulaziz University, Al-Kharj, KSA. The animals were kept under controlled conditions of temperature (22 ± 2 °C), humidity (55%) and light/dark cycle (12/12 h). The animals were provided with Purina chow and free access to drinking water ad libitum [20]. The experimental protocol was approved by the Bioethical Research Committee at Prince Sattam Bin Abdulaziz University.

### 3.6. Hepatoprotective and Nephroprotective Activity

Rats were divided into seven groups of five animals. Group 1 was designated as the control group and received normal saline only. Groups 2–7 received Pa via the intraperitoneal route for two days at 500 mg/kg body weight. Group 2 did not receive any further treatment. Group 3 was treated with 10 mg/kg p.o. (20.7 μmol/kg) of Sil (Sigma-Aldrich, St. Louis, MO, USA) [20]. Groups 4–7 were treated with 20.7 μmole/kg body weight of compounds **2**–**4**, **6**. Treatment started 5 days prior to Pa administration and continued to the end of the experimental protocol. After 24 h, following the second dose of Pa, the animals were sacrificed using ether anesthesia. Blood samples were collected by heart puncture and the serum was separated to estimate the biochemical parameters. The liver and kidney tissues were immediately removed and properly treated for the determination of the tissue parameters. Representative pieces were immersed in 10% formalin for fixation to carry out the histopathological study.

### 3.7. Determination of Biochemical Serum Parameters

Serum glutamate oxaloacetate transaminase (AST), serum glutamate pyruvate transaminase (ALT), gamma glutamyltranspeptidase (GGT), alkaline phosphatase (ALP) and total bilirubin were determined following the reported methods [54]. The enzyme activities were measured by Reflotron^®^ diagnostic strips (Roche, Basel, Switzerland) and a Reflotron^®^ Plus instrument (Roche) (Figure 2 and Appendix A).

Serum creatinine and blood urea were assayed using Randox Diagnostic kits (Randox Laboratories Ltd., Crumlin, U.K.) by the reported method [55]. Potassium levels were measured using diagnostic strips (Reflotron^®^, Roche), while sodium levels were determined photometrically using the Mg-uranyl acetate method (Figure 3 and Appendix A) [56].

### 3.8. Determination of Non-Protein Sulfhydryl Groups and Total Protein

The liver and kidney tissues were homogenized in 0.02 M EDTA using a Potter–Elvehjem type C homogenizer (Sigma-Aldrich, St. Louis, MO, USA). Homogenates equivalent to 100 mg of tissues were used for the measurements. Nonprotein sulfhydryl groups (NP-SH) were quantified by dilution of the homogenates with 4 mL of distilled water and 1 mL of 50% trichloroacetic acid (TCA) followed by shaking for 15 min. The supernatants were then mixed with 2 mL Tris buffer, pH 8.9 and 0.1 mL of 0.01 M of 5,5′-dithio-bis-(2-nitrobenzoic acid) (DTNB). The absorbances were measured spectrophotometrically within 5 min at 412 nm against a reagent blank with no homogenate (Figure 4 and Appendix A) [57]. TP were quantified by mixing homogenates with 0.7 mL of Lowry’s solution followed by incubation in the dark for 20 min at room temperature. Diluted Folin’s reagent (0.1 mL) was then added, and samples were incubated at room temperature in the dark for 30 min. The absorbances were then measured at 750 nm (Figure 5 and Appendix A) [58].

### 3.9. Statistical Analysis

An analysis of variance (ANOVA) test was used to evaluate the significance of the difference between the groups. Non-paired samples such as the control and Pa-treated group were compared for significance using Dunnett’s test [59]. All the reported values are presented as mean ± S.E.

### 3.10. Histopathology

Water-ethanol mixtures with a gradual increase in ethanol were used to dehydrate liver and kidney samples till reaching absolute ethanol, cleared and infiltrated by immersion in increasing concentrations of ethanol (70–100%) and xylene (3 times, 1 h each) followed by paraffin wax (4 times, 1 h each). The tissues were placed by hot forceps in molds, chilled on cold plates and excess waxes were removed. Thin sections of 3 μm were obtained by a rotary microtome (Leitz 1512) and placed onto clean slides. The slides were drained and stained using Mayer’s hematoxylin solution for 15 min after deparaffinization and hydration. The slides were then washed in lukewarm tap water for 15 min then immersed in 80% ethanol for two minutes and counterstained in eosin-phloxine solution for 2 min. The slides were then washed with 95% ethanol, absolute ethanol and xylene (2 min each) and finally mounted in resinous medium [60].

## 4. Conclusions

A phytochemical study of the aerial parts of *Scrophularia hypericifolia* resulted in the isolation of nine acylate iridoid glycosides. Four new compounds were identified as 6′-acetyl hypericifolioside A (**1**) 6′-acetyl hypericifolioside B (**2**) hypericifolioside A (**3**) and hypericifolioside B (**4**). In addition, five known compounds were identified as laterioside (**5**) 8-O-acetylharpagide (**6**) 6-O-L(4″-O-*trans*-cinnamoyl) rhamnopyranosyl catalpol (**7**) lagotisoside D (**8**) and harpagoside (**9**). Compounds **2**–**4** and **6** were tested against paracetamol-induced hepatic and renal toxicity. Compound **3** resulted in 23.26, 33.71, 37.95, 16.67 and 27.70% reductions in the elevated levels of AST, ALT, GGT, ALP and bilirubin. It also reduced the levels of urea, creatinine, sodium and potassium by 24.61%, 36.00%, 33.88% and 35.47%, respectively. Compound **3** restored TP contents and NP-SH in both liver and kidney tissues to levels comparable to Sil. These findings were supported by histopathological study by the almost normal appearance of liver cells, while little degeneration was observed in kidney cells after treatment with **3**. In some of the measured parameters, **3** was as protective as the standard drug Sil. Compounds **4** and **6** expressed less protection, while **2** was almost inactive.

## Figures and Tables

**Figure 1 biology-10-00145-f001:**
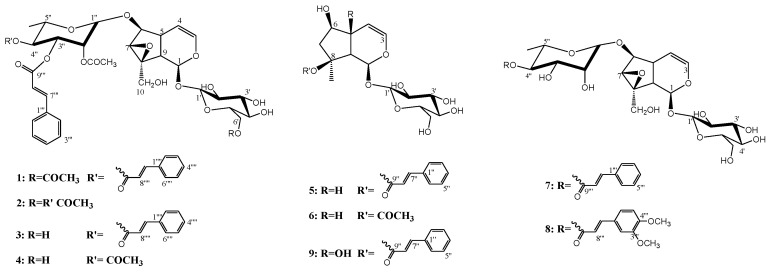
Structures of the isolated compounds (**1**–**9**).

**Figure 2 biology-10-00145-f002:**
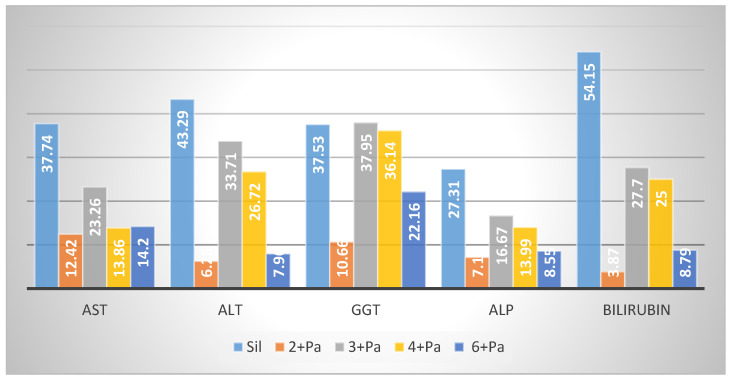
Effect of compounds **2**–**4**, **6** on liver serum biochemical parameters AST, ALT, GGT, APL and bilirubin (% reduction).

**Figure 3 biology-10-00145-f003:**
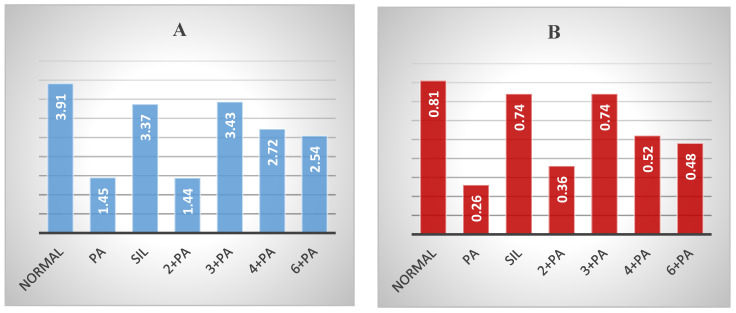
Effect of compounds **2**–**4**, **6** on liver (**A**) and kidney (**B**) TP.

**Figure 4 biology-10-00145-f004:**
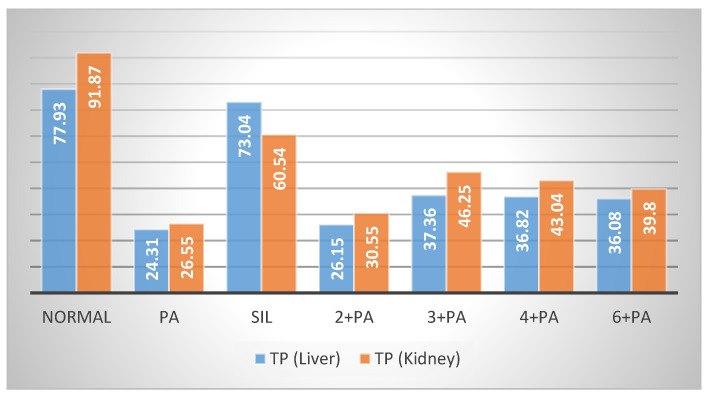
Effect of compounds **2**–**4**, **6** on liver and kidney NP-SH.

**Figure 5 biology-10-00145-f005:**
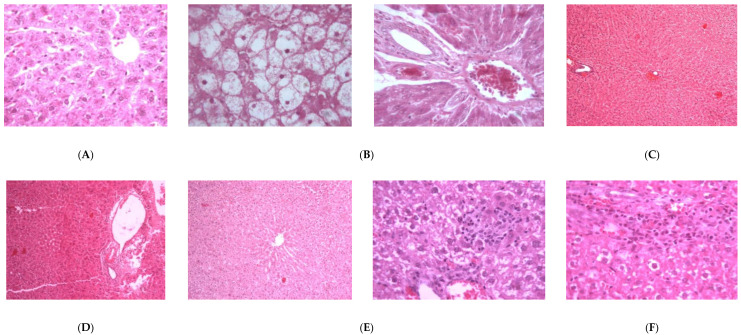
Light microscope pictures of liver tissues: (**A**) Liver cells of normal control group showing central vein, normal sinusoids and cells. (**B**) Liver cells of Pa-treated group showing hydropic swelling, rosette formation, portal dilated congested vessels, focal necrosis, lymphocytic infiltrate and lymphocytic exudate. (**C**) Liver cells of Pa- and Sil-treated group showing mostly normal cells with mild congestion in the central venules. (**D**) Liver cells of group treated with **3** showing almost normal lobules and portal tract with the absence of degeneration and vein congestion. (**E**) Liver cells of group treated with **4** showing mild focal necrosis and portal tract congestion. (**F**) Liver cells of group treated with **6** showing moderate to high degree of degeneration, portal congestion and inflammatory cell infiltration.

**Figure 6 biology-10-00145-f006:**
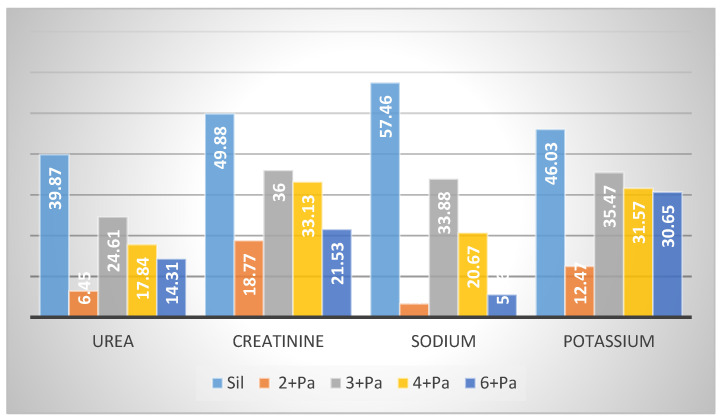
Effect of compounds **2**–**4**, **6** on kidney serum biochemical parameters (% reduction).

**Figure 7 biology-10-00145-f007:**
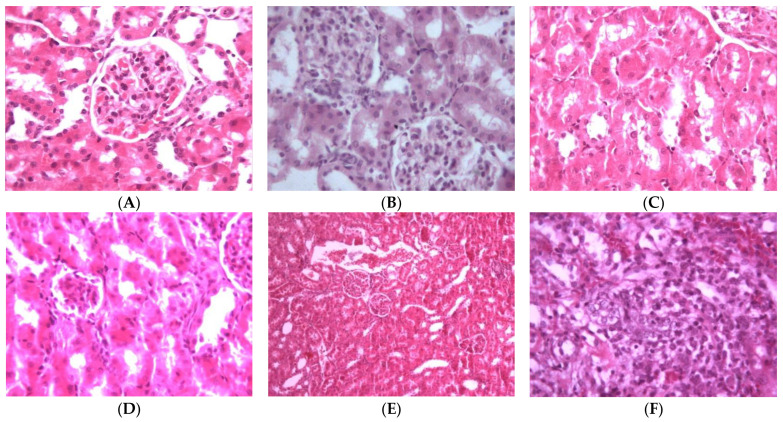
Light microscope pictures of kidney cells: (**A**) Normal cells. (**B**) Kidney cells of Pa-treated group showing abnormal cellular structures with degenerated corpuscle, reduced size of glomeruli, shrinkage of Bowman capsule, cloudy swelling in tubules and inflammatory infiltrate. (**C**) Kidney cells of rats treated with Pa and Sil showing no marked changes, nearly normal glomeruli and mild cloudy swelling in tubules. (**D**) Kidney cells of group treated with **3** showing moderate cortical atrophy and reduced size of glomeruli. (**E**) Kidney cells of group treated with **4** showing kidney cortical congestion, abnormal glomeruli (reduced in size) and cloudy swelling in tubules. (**F**) Kidney cells of group treated with **6** showing inflammatory cell infiltration at corticomedullary junction, kidney cortical infiltration, almost complete absence of Bowman space, degenerated and convoluted necrotic tubules.

**Table 1 biology-10-00145-t001:** ^1^H NMR data (δ ppm, *J* in parentheses in Hz) for compounds 1–4 and 7 *.

Pos.	1	2		3		4		7
	CD_3_OD	DMSO d_6_	Pyridine d_6_	CD_3_OD	Pyridine d_6_	CD_3_OD	Pyridine d_6_	CD_3_OD
1	4.95 (d, 9.8)	4.83 (d, 9.2)	5.45 **	5.14 **	5.51 (d, 8.8)	5.11 **	5.48 **	5.11 **
3	6.43 (d, 5.8)	6.44 (d, 1.5)	6.47 (d, 5.2)	6.43 (bd, 5.5)	6.48 (d, 5.7)	6.42 (bd, 6.0)	6.45 (d, 4.7)	6.41 (d, 5.7)
4	5.16 (t, 5.4)	5.04 **	5.45 **	5.14 **	5.19 (t, 5.0)	5.11 **	5.12 **	5.11 **
5	2.54 (bq, 4.5)	2.39 (bs)	2.76 (bs)	2.55 (bq, 4.3)	2.76 (bq, 3.9)	2.51 (bq, 4.6)	2.73 (bs)	2.45 (bq, 4.4)
6	4.03 (d, 8.0)	3.93 **	4.25 **	4.11 (d, 8.0)	4.12 **	4.09 (d, 8.4)	4.07 (d, 7.6)	4.05 (d, 8.1)
7	3.71 (s)	3.69 (s)	3.93 (s)	3.72 (s)	3.80 (s)	3.70 (s)	3.77 (s)	3.69 (s)
9	2.65 (t, 9.0)	2.46 (bt, 8.0)	2.85 (t, 7.7)	2.64 (t, 9.4)	2.84 (t, 8.9)	2.62 (bt, 9.3)	2.82 (bt, 8.0)	2.60 (t, 8.0)
10	3.78 (d, 13.0)	3.63 (d, 12.4)	4.39 (d, 13.0)	3.87 (d, 13.2)	4.48 **	3.85 (d, 13.2)	4.47 (s)	3.85 **
	4.18 **	3.93 **	4.50 (d, 13.0)	4.16 (m)		4.18 (d, 13.2)		4.18 (d, 13.2)
1′	4.83 (d, 7.9)	4.66 (d, 7.0)	5.45 (Overl.)	4.83 (d, 7.9)	5.51 (d, 8.8)	4.80 (d, 7.9)	5.48 **	4.80 (d, 7.9)
2′	3.32 (m)	3.11 (bt, 7.5)	4.10 (m)	3.32 (m)	4.12 **	3.28 (m)	4.14 (t, 7.8)	3.30 (m)
3′	3.43 (m)	3.25 (bt, 8.4)	4.25 **	3.46 (t, 9.0)	4.30 **	3.44 (bt,9.0)	4.07 (d, 7.6)	3.43 (q, 8.9)
4′	3.43 (m)	3.19 (bt, 8.9)	4.10 (m)	3.32 (m)	4.19 (bt, 9.2)	3.28 (m)	4.20 (t, 8.7)	3.30 (m)
5′	3.52 (m)	3.42 (m)	3.99 (m)	3.37 (m)	4.01 (bt, 6.7)	3.33 (m)	4.00 (bs)	3.30 (m)
6′	4.28 (dd, 5.0, 12.0)	4.14 (m)	4.69 (dd, 3.7, 11.4)	3.67 (dd, 6.5, 11.9)	4.30 **	3.65 (dd, 6.8, 12.0)	4.30 **	3.67 **
	4.47 (d, 6.8)	4.31 (m)	4.88 (d, 11.4)	3.96 (d, 10.9)	4.53 (d, 11.8)	3.94 (bd, 12.0)	4.53 (d, 10.7)	3.93 **
1″	5.13 (s)	5.14 (s)	5.45 **	5.14 **	5.43 (s)	5.11 **	5.38 (s)	5.03 (s)
2″	5.43 (bs)	5.27 (bs)	5.91(s)	5.43 (bs)	5.94 **	5.39 (m)	5.88 (s)	3.96 (m)
3″	5.53 (dd, 2.7,10.0)	5.27 (bs)	5.95 (d, 7.7)	5.55 (dd, 3.2, 10.2)	6.09 (dd, 2.6, 10.0)	5.41 (m)	5.94 (bd, 9.7)	3.96 (m)
4″	5.32 (t, 10.0)	5.04 **	5.71 (t, 9.6)	5.33 (t, 10.0)	5.89 **	5.17 (t, 9.8)	5.71 (t, 9.7)	5.11 **
5″	4.18 **	3.93 **	4.34 (m)	4.16 **	4.44 (m)	4.06 (m)	4.30 **	3.67 **
6″	1.27 (d, 6.0)	1.16 (d, 4.3)	1.38(d, 5.5)	1.28 (d, 6.2)	1.46(d, 6.0)	1.24 (d, 6.3)	1.37 (d, 5.4)	1.20 (d, 6.4)
3‴–5‴	7.35 **	7.43 (bs),	7.30 (bs),	7.34 **	7.31 (bs)	7.43 (m, 3H),	7.20- 7.57 **	7.43 (bs)
2‴, 6‴	7.49 (d, 6.6)	7.69 (bs)	7.47 (d, 5.8)	7.47 (d, 6.4)	7.24 (d, 7.3)	7.61 (m, 2H)		7.64 (bs)
7‴	7.61 (d, 16.0)	7.60 (d, 16.0)	7.91 (d, 16.0)	7.60 (d, 16.0)	7.89 (d, 16.0)	7.67 (d, 16.0)	7.92 (d, 16.0)	7.76 (d, 16.0)
8‴	6.40 (d, 16.0)	6.54 (d, 16.0)	6.74 (d, 16.0)	6.39 (d, 16.0)	6.72 (d, 16.0)	6.46 (d, 16.0)	6.74 (d, 16.0)	6.61 (d, 16.0)
3″″–5″″	7.35 **	-	-	7.34 **	7.57(bs)	-	-	-
2″″, 6″″	7.53 (d, 6.6)			7.51 (d, 6.1)	7.39 (d, 7.2)			
7″″	7.70 (d, 16.0)	-	-	7.71 (d, 16.0)	7.99 (d, 16.0)	-	-	-
8″″	6.50 (d, 16.0)	-	-	6.49 (d, 16.0)	6.80 (d, 16.0)	-	-	-
Acetyl:								-
6′	2.07 (s)	2.00 (s)	1.94 (s)	-	-	-	-	
2″	2.19 (s)	2.02 (s)	2.06 (s)	2.19	2.10 (s)	2.06 (s)	2.07 (s)	
4″	-	2.13 (s)	2.11 (s)	-	-	2.18 (s)	2.10 (s)	

* Assignments based on DEPT135, COSY, HSQC, HMBC, H2BC, NOESY experiments and comparison with reported data. ** Overlapped signals.

**Table 2 biology-10-00145-t002:** ^13^C NMR data (δ ppm) for compounds 1–4 and **7** *.

Pos.	1 **	2 ***	3 **	4 **	7 **	Pos.	1 **	2 ***	3 **	4 **	7 **
1	93.95	94.03	93.82	93.76	93.83	3″	69.45	69.22	69.46	69.39	68.84
3	141.14	141.75	141.14	141.12	140.97	4″	71.05	70.70	71.07	70.90	74.15
4	102.06	102.46	101.95	101.80	102.09	5″	66.89	65.89	66.88	66.65	66.86
5	35.78	35.80	35.81	35.75	35.93	6″	16.53	17.62	16.54	16.35	16.54
6	83.74	83.21	83.61	83.47	82.77	1‴	134.00	134.11	133.97	134.07	134.32
7	58.02	57.56	58.14	58.07	58.13	2‴–6‴	128.01–130.45	129.01–131.26	128.00–130.43	128.04–130.46	127.95–130.26
8	65.08	66.62	65.21	65.19	65.25	7‴	146.28	146.07	146.26	146.04	145.39
9	41.85	42.14	41.92	41.87	41.91	8‴	116.50	117.41	116.51	116.47	117.42
10	60.17	59.22	60.07	60.01	60.06	9‴	165.93	165.75	165.92	165.89	166.98
1′	98.45	98.71	98.36	98.32	98.32	1″″	134.00	-	133.97	-	-
2′	73.35	73.70	73.44	73.42	73.42	2″″–6″″	128.01–130.45	-	128.00–130.43	-	-
3′	76.02	76.51	76.28	76.27	76.25	7″″	146.07	-	146.05	-	-
4′	70.06	70.26	70.37	70.37	70.35	8″″	116.50	-	116.51	-	-
5′	74.36	74.22	77.22	77.22	77.19	9″″	166.31	-	166.30	-	-
6′	62.74	63.27	61.58	61.55	61.52	Acetyl: 6′	19.54, 171.40	20.94, 170.72	-		-
1″	96.41	96.11	96.41	96.28	99.06	2″	19.48, 170.32	21.07, 170.23	19.49, 170.31	19.37, 170.30	-
2″	69.92	69.55	70.11	69.92	71.05	4″	-	21.04, 170.06	-	19.39, 170.53	-

* Assignments based on DEPT135, COSY, HSQC, HMBC, H2BC, NOESY experiments and comparison with reported data. ** Spectra measured in CD_3_OD. *** Spectra measured in DMSO d_6._

## Data Availability

The data presented in this study are available in Appendix A.

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
