# Peer review of "In Vivo Hepatoprotective and Nephroprotective Activity of Acylated Iridoid Glycosides from Scrophularia hepericifolia"

_biology, 2021, doi:10.3390/biology10020145_

Round 1

Reviewer 1 Report

The present work, “In vivo hepatoprotective and nephroprotective activity of acylated iridoid glycosides from Scrophularia hypericifolia against high paracetamol doses induced toxicity”, describes the isolation and the structural determination of four new acylated iridoid glycosides, together with five known iridoids. The structures were elucidated on the basis of spectroscopic data (NMR and MS). Pharmacological analysis demonstrated that some of these compounds showed in vivo hepatoprotective and nephroprotective activities.

The novelty and originality of the paper is fair because it describes the structural characterization of four new metabolites from Scrophularia hypericifolia, and it reports a deep in vivo pharmacological evaluation. These features may justify its publication on Biology but after major revisions.

The determination of the planar structures requires deep revision, whereas the assignment of stereochemistry appears totally absent (despite NOESY and ROESY are present in the supplementary). Data presentation can most likely be addressed in a revised version. 

As concern the pharmacological data, Scrophularia hypericifolia extracts as well as the enriched fractions were also explored? Is it possible identified a putative biological target, responsible of the in vivo hepatoprotective and nephroprotective activities showed by the tested compounds?

In addition, there are many minor revisions required:

Page    Line                Comments

1          2                      The name of the plant in italics

1          3                      The title could be changed as “In vivo hepatoprotective and nephroprotective activity of acylated iridoid glycosides from Scrophularia hepericifolia

1          12                    The name of the plant in italics

1          12                    The abstract could be improved and rephrased

2          77                    The number of compounds have to be in bold.

3          90                    The compounds 1-4 could be compared to compound 7, with the same aglycone moiety and sugar units (a rhamnose unit at C-6 and glucose unit at C-1). I suggest to add compound 7 in tables 1 and 2.

3          91-96               I suggest to change the phrase with “The presence of rhamnose moiety at C-6 in compounds 1-4 was confirmed by the presence of methyl protons (doublets, at dH 1.27, 1.16, 1.28 and 1.24 for compounds 1,2, 3 and 4) in 1H NMR spectra, correlated with the……

3          97                    The number of compounds have to be in bold.

4          102                  Table1 and 2  If possible, you could list the values of dH and dC for positions 2’’’-6’’’ and 2’’’’ and 6’’’’

5          108                  [M+Na]+ [M+K]+

5          108-170           The structural characterization should have changed: I suggest to describe before compounds 4 and 2 that compared to compound 7 are characterized by the presence of one or two additional acetyl moieties (from mass data). Compounds 3 and 1 could be described compared to compounds 4 and 2, from which they differ for the presence of an additional trans-cinnamoyl moiety at C2’’.

            191                  Sil is for silymarin. This abbreviation appears for the first time in the text

            250                  The name of the plant in italics

            297                  The name of the plant in italics

            313-330           The number of compounds have to be in bold.

            334                  [M+K]+

            338                  [M-H]-

            342                  [M+K]+

            345                  [M+K]+

            346                  [M-H]-

            412                  The name of the plant in italics

            416                  @ symbol

            418- 421          The number of compounds have to be in bold.

Author Response

As authors of this manuscript we would like to express our indispensable great appreciations to the reviewers for their valuable and constructive comments. Applying reviewer’s advice along with collection of few more NMR data enable the correction of the structures with much more solid evidence. Their comments really increase the quality of our data presentation. We really appreciate this great job.

First Reviewer

Comments and Suggestions for Authors

The determination of the planar structures requires deep revision, whereas the assignment of stereochemistry appears totally absent (despite NOESY and ROESY are present in the supplementary). Data presentation can most likely be addressed in a revised version. 

Since the basic skeleton of the compounds is well known we just compared our data with similar compounds reported in Ref 36. We added brief discussion and used NOESY data to support the stereochemistry of the basic skeleton. It is already well established in the literature. More NOESY were added to supplementary materials.

As concern the pharmacological data, Scrophularia hypericifolia extracts as well as the enriched fractions were also explored? Is it possible identified a putative biological target, responsible of the in vivo hepatoprotective and nephroprotective activities showed by the tested compounds?

A part was added to the discussion and one relevant reference was cited.

In addition, there are many minor revisions required:

Page    Line                Comments

1          2                      The name of the plant in italics

Corrected

1          3                      The title could be changed as “In vivo hepatoprotective and nephroprotective activity of acylated iridoid glycosides from Scrophularia hepericifolia

Title changed as suggested.

1          12                    The name of the plant in italics

Scientific plant names were checked throughout the manuscript and changed to italics.

1          12                    The abstract could be improved and rephrased

Revision of the abstract was performed.

2          77                    The number of compounds have to be in bold.

Changed to bold.

3          90                    The compounds 1-4 could be compared to compound 7, with the same aglycone moiety and sugar units (a rhamnose unit at C-6 and glucose unit at C-1). I suggest to add compound in tables 1 and 2.

Data of 7 were added to Tables 1 and 2.

3          91-96               I suggest to change the phrase with “The presence of rhamnose moiety at C-6 in compounds 1-4 was confirmed by the presence of methyl protons (doublets, at dH 1.27, 1.16, 1.28 and 1.24 for compounds 1,23 and 4) in 1H NMR spectra, correlated with the……

Suggested changes were undertaken.

3          97                    The number of compounds have to be in bold.

Changed to bold.

4          102                  Table1 and 2  If possible, you could list the values of dH and dC for positions 2’’’-6’’’ and 2’’’’ and 6’’’’

We did our best to assign the protons and carbons, However, some values were very difficult to firmly assign due to the overlapping.

5          108                  [M+Na][M+K]+

The whole manuscript was check for the proper format of Mass ions.

5          108-170           The structural characterization should have changed: I suggest to describe before compounds 4 and 2 that compared to compound 7 are characterized by the presence of one or two additional acetyl moieties (from mass data). Compounds and 1 could be described compared to compounds 4 and 2, from which they differ for the presence of an additional trans-cinnamoyl moiety at C2’’.

Based on the comment and the other reviewer comments the part concerning structure elucidation was subjected to extensive rearrangement and revision.

            191                  Sil is for silymarin. This abbreviation appears for the first time in the text

Abbreviation was induced after the first use in the introduction.

            250                  The name of the plant in italics

Scientific plant names were checked throughout the manuscript and changed to italics.

            297                  The name of the plant in italics

Scientific plant names were checked throughout the manuscript and changed to italics.

            313-330           The number of compounds have to be in bold.

All necessary Changes were undertaken.

            334                  [M+K]+

The whole manuscript was check for the proper format of Mass ions.

            338                  [M-H]-

The whole manuscript was check for the proper format of Mass ions.

            342                  [M+K]+

The whole manuscript was check for the proper format of Mass ions.

            345                  [M+K]+

The whole manuscript was check for the proper format of Mass ions.

            346                  [M-H]-

The whole manuscript was check for the proper format of Mass ions.

            412                  The name of the plant in italics

Scientific plant names were checked throughout the manuscript and changed to italics.

            416                  @ symbol

Correct to the proper font.

            418- 421          The number of compounds have to be in bold.

All necessary Changes were undertaken.

Reviewer 2 Report

The paper reports the isolation, from the aerial parts of the plant Scrophularia hypericifolia, of four unreported iridoid glycosides together with five know related compounds. The compounds isolated in enough amount were also subjected to in vivo evaluation for their ability to reduce the liver and kidney injury induced by high paracetamol doses.

The paper is highly professional in quality as concerning the reported MS and NMR spectra . However, in my opinion the description in the main text of the characterization analysis is very confused and difficult to the readers. As usual, the structure of one parent compound, for instance 1, should be discussed in detail, starting from each subunit, followed by their connectivity. Then the structures of related compound should be assigned by comparison of the spectral data with the parent compound (already done but to be improved). Therefore I suggest to totally revise this part.

There are some discrepancies in the proton scalar pattern Table 1. Considering that JAB=JBA some vicinal constant values appear not coherent (i.e. H-9 t, 9.0, H-5 bq, 4.5, H-1 d, 9.8).

In my opinion when possible please use the scalar couplings by HMBC rather the dipolar couplings by NOESY to connect the subunits. For instance the assignment stated in the sentence (lines 124-126): The cinnamoyl moiety with the two doublets at dH 6.40, 7.61 (each d, J=16 Hz) showed NOE correlations with H-6’’ at dH 1.27 (d, J= 6 Hz) and 125 was assigned to position 4’’ of rhamnose while the other cinnnamoly (to be corrected to cinnamoyl) with the two doublets at dH 6.50, 7.70 (each d, J=16 Hz) located at position 2’’ raised some doubts since H-6’’ is located from the same face of -OR in 2’, whereas it is from the opposite face of -OR in 4’. As matter of fact, the related part of the NOESY spectrum is nor reported in the supplementary material.

Also Figure 7 is missing.

Author Response

As authors of this manuscript we would like to express our indispensable great appreciations to the reviewers for their valuable and constructive comments. Applying reviewer’s advice along with collection of few more NMR data enable the correction of the structures with much more solid evidence. Their comments really increase the quality of our data presentation. We really appreciate this great job.

Second Reviewer

Comments and Suggestions for Authors

The paper reports the isolation, from the aerial parts of the plant Scrophularia hypericifolia, of four unreported iridoid glycosides together with five know related compounds. The compounds isolated in enough amount were also subjected to in vivo evaluation for their ability to reduce the liver and kidney injury induced by high paracetamol doses.

The paper is highly professional in quality as concerning the reported MS and NMR spectra . However, in my opinion the description in the main text of the characterization analysis is very confused and difficult to the readers.

As usual, the structure of one parent compound, for instance 1, should be discussed in detail, starting from each subunit, followed by their connectivity. Then the structures of related compound should be assigned by comparison of the spectral data with the parent compound (already done but to be improved). Therefore I suggest to totally revise this part.

The discussion was revised as directed with some improvements.

There are some discrepancies in the proton scalar pattern Table 1. Considering that JAB=JBA some vicinal constant values appear not coherent (i.e. H-9 t, 9.0, H-5 bq, 4.5, H-1 d, 9.8).

The J values for H-9 and H-1 are consistent with the reported data. Kindly see Pharmaceutical Biology, 2011; 49(11): 1150–1157 cited as Ref. 36. H-5 usually reported as mutiplet, however, in our data it could be presented as broad quartet. We can change it to multiplet if you see that will be more appropriate.

In my opinion when possible please use the scalar couplings by HMBC rather the dipolar couplings by NOESY to connect the subunits.

That suggestion was very effective and interesting. HMBC correlations were extensively included in the discussion and more figures were added to the supplementary materials. More HMBC experiments for compounds 2 and 4 were performed. Deep analyses of the HMBC data enable to draw solid assignment for the substitutions positions. Appropriate changes were made and the four structures were revised. Again we appreciate you valuable comment.

For instance the assignment stated in the sentence (lines 124-126): The cinnamoyl moiety with the two doublets at dH 6.40, 7.61 (each d, J=16 Hz) showed NOE correlations with H-6’’ at dH 1.27 (d, J= 6 Hz) and 125 was assigned to position 4’’ of rhamnose while the other cinnnamoly (to be corrected to cinnamoyl) with the two doublets at dH 6.50, 7.70 (each d, J=16 Hz) located at position 2’’ raised some doubts since H-6’’ is located from the same face of -OR in 2’, whereas it is from the opposite face of -OR in 4’. As matter of fact, the related part of the NOESY spectrum is nor reported in the supplementary material.

Solid evidences for the positions assignments were obtained from HMBC as suggested. Doubts regarding NOE use were true and structures were corrected. Spelling mistake was corrected. More HMBC expansions were added to supplementary materials.

Also Figure 7 is missing.

The missing figure is included.

Round 2

Reviewer 1 Report

The authors improved the structural description of new compounds 1-4. Now the manuscript is appropriate for publication in Biology.

There are other minor revisions required. In the structural elucidation of compound 1, I suggest starting in the description of mass data followed by 1H NMR data and then  13C assignment.

 The authors must correct the figure 1 since the drawing of the rhamnose unit does not follow the correct notation. The last figure in the SI reporting H2BC correlations of the rhamnosyl moiety of 3 should be changed with round arrows to indicate the HMBC correlations and indicating the  substituted positions of the sugar unit. 

ppm could be omitted if it is reported δC or δH (for examples line 94 in page 3 ect.).

Author Response

We would like to thank the reviewer’s nice comments.

There are other minor revisions required. In the structural elucidation of compound 1, I suggest starting in the description of mass data followed by 1H NMR data and then  13C assignment.

Rearrangement of this part was done.

 The authors must correct the figure 1 since the drawing of the rhamnose unit does not follow the correct notation.

Necessary changes were performed.

The last figure in the SI reporting H2BC correlations of the rhamnosyl moiety of should be changed with round arrows to indicate the HMBC correlations and indicating the  substituted positions of the sugar unit. 

HMBC correlation were included and the figure was improved with some colours.

ppm could be omitted if it is reported δC or δH (for examples line 94 in page 3 ect.).

The abbreviation “ppm” was deleted as directed when appropriate.

Reviewer 2 Report

The revised version of paper appears now acceptable, both for the clarity of the presentation and for the soundness of the structural assignments. Some discrepancies in 3J coupling analysis remain: for instance H 4’’ In compound 3 recorded in pyridine is clearly a triplet: the last arm overlaps with the bs at 5.94 as readily seen in 2nd and 3rd extension of the H2SC spectrum in the SI. Again, the signal at dH 5.52 (compound 3 in pyridine) is a doublet.  Please revise again using a scientific approach: a d with J=9.8 cannot became a singlet simply by solvent change!

Overl. is not standard. The use of a footnote to indicate overlapped signals would be okay.

Author Response

Response to Reviewers Round 2:

The revised version of paper appears now acceptable, both for the clarity of the presentation and for the soundness of the structural assignments.

We express our thanks to the reviewer’s valuable comments. It actually help to increase the manuscript value.

Some discrepancies in 3J coupling analysis remain: for instance H 4’’ In compound 3 recorded in pyridine is clearly a triplet: the last arm overlaps with the bs at 5.94 as readily seen in 2nd and 3rd extension of the H2SC spectrum in the SI.

The description of these two protons changed to “overlapped”. The added proton expansion revealed that this description is more appropriate.

Again, the signal at dH 5.52 (compound 3 in pyridine) is a doublet.  Please revise again using a scientific approach: a d with J=9.8 cannot became a singlet simply by solvent change!

Expansion of the HSQC and COSY (added to supplementary figures) make it possible to describe this signal as doublet with chemical shift 5.51 and J=8.8 Hz integrated for the two protons 1 and 1’.

Overl. is not standard. The use of a footnote to indicate overlapped signals would be okay.

Changes made as directed.
